# Multilingual Hate Speech Detection: A Semi-Supervised Generative Adversarial Approach

**DOI:** 10.3390/e26040344

**Published:** 2024-04-18

**Authors:** Khouloud Mnassri, Reza Farahbakhsh, Noel Crespi

**Affiliations:** Samovar, Télécom SudParis, Institut Polytechnique de Paris, 91120 Palaiseau, France

**Keywords:** social media, hate speech, semisupervised, GAN, multilingual, PLMs

## Abstract

Social media platforms have surpassed cultural and linguistic boundaries, thus enabling online communication worldwide. However, the expanded use of various languages has intensified the challenge of online detection of hate speech content. Despite the release of multiple Natural Language Processing (NLP) solutions implementing cutting-edge machine learning techniques, the scarcity of data, especially labeled data, remains a considerable obstacle, which further requires the use of semisupervised approaches along with Generative Artificial Intelligence (Generative AI) techniques. This paper introduces an innovative approach, a multilingual semisupervised model combining Generative Adversarial Networks (GANs) and Pretrained Language Models (PLMs), more precisely mBERT and XLM-RoBERTa. Our approach proves its effectiveness in the detection of hate speech and offensive language in Indo-European languages (in English, German, and Hindi) when employing only 20% annotated data from the HASOC2019 dataset, thereby presenting significantly high performances in each of multilingual, zero-shot crosslingual, and monolingual training scenarios. Our study provides a robust mBERT-based semisupervised GAN model (SS-GAN-mBERT) that outperformed the XLM-RoBERTa-based model (SS-GAN-XLM) and reached an average F1 score boost of 9.23% and an accuracy increase of 5.75% over the baseline semisupervised mBERT model.

## 1. Introduction

Generative Artificial Intelligence (Generative AI) has fundamentally revolutionized the field of Natural Language Processing (NLP), thus adding outstanding changes in text summarization, translation, classification, and of course text generation tasks. One of the major reasons for this paradigm transformation is the release of large-scale models like Generative Adversarial Networks (GANs) and GPT. For example, GPT-3 has demonstrated remarkable text generation abilities across different NLP tasks, including storytelling and coding [1]. Additionally, generative models like XLM-RoBERTa or mBERT have also participated in advancing machine translation techniques [2]. Moreover, using generative AI models for data augmentation and semisupervised learning has constructed more robust models, thus reducing the need for labeled data [3]. Getting deeper into how far Generative AI can go, it has proven its capacity to generate social media-like content and also to annotate it [4].

In recent years, social media platforms like Facebook and Twitter have become more and more famous and widely used for connecting and communicating. These platforms contribute enormously to creating bridges between different countries and cultures, thus illustrating multiculturalism and multilingualism [5]. Even though the freedom to communicate and express opinions is one of the noteworthy aspects on social media, this privilege is often misused and serves as a means for disseminating hate speech and offensive content online [6]. An increasing consideration has been shown that many users have reported encountering hate speech and offensive content on these platforms [7]. In fact, due to the anonymity delivered on social media, users are becoming more free to express themselves and more likely to be engaged in hateful actions [8].

In order to give a detailed overview of the concept, we aim to illustrate the definition of hate speech on social media. Hate speech is a specific subset of offensive language that directly targets individuals or groups based on specific features with the intent to discriminate or incite harm [9]. Generally, hate speech is defined as a conscious and intended public expression aimed at criticizing a specific group of people, whether based on race, religion, ethnicity, nationality, gender, sexual identity, or orientation. Recently, the emergence of social media platforms has intensified the spread of this content. And defining hate speech in this field becomes challenging due to its different forms of expression, including symbolic, verbal, nonverbal, etc. [10]. Moreover, online hate speech usually uses imprecise or metaphorical language, thereby making it more difficult to determine or to build a unique standard definition to be used worldwide. Particularly, it can be considered sometimes as socially acceptable to express negative stereotyping [10]. Overall, we provide a definition of this phenomenon in a survey paper in which we established extensive study, and we define hate speech as any content that targets individuals or groups based on several factors such as race, ethnicity, religion, sexual orientation, gender, or other identifiable characteristics. This concept often reflects the policies and guidelines set by multiple social media platforms, which are influenced by legal frameworks and societal standards [9]. In addition, the spread of this content exceeds linguistic borders and encompasses more languages over time. Consequently, there is a crucial need to restrain this viral spread, especially since it can lead to severe crimes against minorities or vulnerable groups [11].

In the beginning, efforts to moderate the spread of hate speech on social media depended on strategies like keyword filters and crowdsourcing, along with human moderators who check flagged content to define if it is considered as hateful or not. While these manual techniques helped in this field, they still require lot of effort, time, and money, especially with the challenges faced by the growing volume of this content spread online. As a result, it becomes more and more difficult to manually moderate it. Therefore, there have been several initiatives to automate the multilingual detection of hateful content, which remains a challenging task [11]. Among the most common challenges, is the cultural backgrounds, which affect the interpretation of this content, that impact its perception across various regions and populations, even within the same language. This complexity is made by the various dialects within languages like Arabic [9]. Moreover, users are becoming more familiar with the automatic detection algorithms, and they have discovered many ways to censor their hateful content to prevent its detection. For example, there is the likely manipulation of words, such as substituting letters with visually equivalent numbers (e.g., replacing “l” with “1” or “E” with “3”) [12]. Another example is illustrated in this research paper [13], which analyzes the Israeli–Palestinian conflict on TikTok and demonstrates how users try to avoid censorship by manipulating their language.

Most of the existing machine learning solutions (monolingual and multilingual) have used supervised learning approaches [9,11], where transfer learning techniques, based on Pretrained Language Models (PLMs), have proven to give outstanding results in multilingual hate speech detection. In fact, transformer-based architectures, such as BERT [14], have been demonstrated to achieve state-of-the-art performance in a variety of hate speech detection tasks. As a result, a large number of BERT-based approaches have been presented in this field [15,16,17,18] etc. Moreover, multilingual transformers, particularly mBERT (multilingual BERT) or XLM-RoBERTa, have been implemented in the multilingual domain for hate speech detection tasks. These models have provided cutting-edge performance in crosslingual and multilingual settings, where several studies demonstrate their usefulness in many languages, especially in low-resource ones [19,20].

While supervised NLP text classification approaches have made impressive advances, they still encounter difficulties in obtaining enough annotated data, which is further complicated in multilingual sentiment analysis tasks like hate speech detection. More specifically, acquiring such high-quality labeled corpora is expensive and time-consuming [12]. Adding to that, multilingual robust models often depend on rich linguistic resources, which are mostly available in English (as a resource-rich language). As a result, these models meet generalization issues that yield decreased performance when used with low-resource languages [21]. As a solution for these deficiencies, semisupervised SS-Learning was introduced in order to reduce the necessity for labeled data. It helps building efficient models that are able to use unlabeled corpora while utilizing only small-sized annotated samples. Thus, SS-Learning was largely used in NLP for hate speech detection tasks [22,23]. One of these SS techniques is the Generative Adversarial Network (GAN) [24], which is based on an adversarial process, where a “discriminator” learns to distinguish between real and generated instances produced by a “generator” that simulates data based on a specific distribution. An extension of GANs is semisupervised SS-GANs, where the “discriminator” also classifies and assigns a class to each data sample [25]. It becomes a remarkable solution in semisupervised learning in hate speech detection, which has been widely used in combination with pretrained language models like SS-GAN-BERT [26] (non-English language).

In this paper, we extended our previous work [27] by proposing a semisupervised generative adversarial framework, in which we include PLMs (mBERT and XLM-RoBERTa) for multilingual hate speech and offensive language detection. This approach leverages the PLM’s capacity to generate high-quality text representations and to adjust to nonannotated data, thus contributing to enhancing the GAN’s generalization for hate speech detection in multiple languages. Even though GAN-BERT has been utilized for different non-English languages in NLP, the semisupervised GAN-PLM approach remains underexplored, especially in multilingual hate speech detection tasks. Therefore, this study aims to fill this gap by proposing the SS-GAN-PLM model for hate speech and offensive language detection across English, German, and Hindi. The key contributions are as follows:Using mBERT [27] and XLM-RoBERTa, we proposed a model, namely SS-GAN-PLM, in multilingual and zero-shot crosslingual settings, and we compared it with the baseline semisupervised mBERT, as well as investigated and compared the capacity of Pretrained Language Models (PLMs) within a generative adversarial framework to enhance sentiment analysis tasks across diverse linguistic contexts.We trained our proposed models across three paradigms: multilingual, crosslingual (zero-shot learning), and monolingual, thereby aiming to examine linguistic feature sharing within Indo-European languages, and we demonstrated their crucial role in enhancing text classification tasks.We explored SS-GAN-PLM’s progressive influence in improving performance through iterative labeled data increases in a multilingual scenario, thus delving into the extent to which the models can perform independently of labeled data.

## 2. Literature Review

### 2.1. GAN for Hate Speech Detection

Generative AI data augmentation is a strategy that applies modifications to a dataset to improve its size and its diversity. Usually, this technique is especially helpful in classes with small sample sizes, since it balances the dataset and enhances model generalization. By producing synthetic data, data augmentation reduces class imbalances, helps avoid overfitting, and improves model performance [28]. In this context, Cao et al. [29] (2020) developed HateGAN, a deep generative reinforcement learning network aimed at augmenting datasets including hateful tweets. HateGAN is built on reinforcement learning and takes influence from Yu et al.’s study on SeqGAN [30] (2017). Their work introduced a gradient reward policy that controls the generation function, thus encouraging the generator to produce more realistic English samples. Their research analyzed text’s hatefulness across six hateful content dimensions by integrating a pretrained toxicity scorer as a multilabel classification model. The production of hateful material is directed by these toxicity scores, which act as feedback signals. These scores are used as rewards to modify the generator’s parameters, which eventually helps to produce more realistic hateful content. The HateGAN model was trained using a policy gradient method to overcome sequence generation issues, and its outcomes highlighted an improvement in the precision of identifying hate speech. Although the use of reinforcement learning is valued, the authors did not show obvious improvements in outcomes or offer a thorough explanation of how it affected the model’s performance. Therefore, at this point, we chose not to apply this methodology to our strategy.

### 2.2. GAN-PLM

Aiming to overcome the time-consuming and expensive labeling process, semisupervised learning has drawn increasing attention as a viable solution. This approach seeks to achieve equivalent or even better performance than supervised algorithms by employing both labeled and unlabeled data. A famous technique in this domain is the use of Generative Adversarial Networks (GANs) [24], which utilize a discriminator to distinguish between generated and real data and a generator to generate synthetic text samples. In fact, GANs have proven their ability to improve the generalization and robustness of text classification models and pretrained language models like BERT, thereby allowing them to efficiently use unlabeled data [31].

In this context, GAN-BERT was first introduced by Croce et al. [31] (2020) as a viable solution to deal with the lack of annotated data. They presented a novel method that uses unlabeled data in a generative adversarial framework to extend the BERT fine-tuning process. Their approach achieved impressive performance across several text classification tasks with as little as 50–100 annotated examples, thus significantly reducing the need for annotated data. Their GAN-BERT model integreated a semisupervised GAN model into a fine-tuned BERT model, where a generator generates synthetic samples that imitate the real data distribution, and BERT operates as the discriminator. This hybrid method makes use of unlabeled data to enhance the model’s generalization capabilities while leveraging BERT’s capacity to produce high-quality representations of input texts. Furthermore, their evaluation tests consistently revealed that GAN-BERT improves the robustness of the model without adding inference cost, because the generator is only used for training, and the discriminator is only used for inference.

Numerous studies were inspired by this model’s outstanding results, and numerous approaches were developed for various tasks. In 2022, Cho et al. [32] presented Linguistically Informed Semi-Supervised GAN with Multiple Generators (LMGAN), a novel approach to semisupervised learning. Their model makes use of BERT’s hidden layers and includes several generators instead of a single one. More specifically, they used the linguistically meaningful intermediate hidden layer outputs of BERT to enhance fake data distribution. Using the hidden layers of BERT (instead of only the last layer) and a basic generator, they managed to improve the quality of the generated data. In fact, when a final generator uses BERT’s embeddings from the GAN-BERT model, it transmits information about real data distribution that would mislead the trained discriminator. Therefore, to apply richer representations of generated data, LMGAN uses numerous generators and the linguistically relevant hidden layers of BERT. Displaying the results of BERT’s hidden layers, they confirmed the significance of having multiple generators, with up to 1.8% improvements in the results. Moreover, Auti et al. [33] utilized the GAN-BERT model for pharmaceuticals text classification tasks. Trained exclusively on biomedical data, GAN-BioBERT [34] gave the best-performing results.

In 2023, Jain et al. [35] introduced the GAN-BERT model with consumer sentiment analysis aspect fusion, which adds semisupervised adversarial learning to enhance the BERT model’s fine-tuning performance. They took different service elements out of customer evaluations and combined them with the word sequences before adding them to the model. The accuracy of the provided model was demonstrated by their examination of the results and their comparison with other models that have been found in earlier work. Adding to that, Du et al. [36] presented a novel approach for job recommendation tasks with Large Language Models (LLMs). They went beyond users’ self-descriptions to extract both explicit features and implicit traits derived from their behaviors, thereby improving the accuracy of user profiling for resume filling. They offered an approach called LGIR (LLM-based GANs Interactive Recommendation) that uses Generative Adversarial Networks (GANs) along with ChatGLM-6B to align unpaired low-quality resumes with high-quality resumes in order to address this problem. Moreover, Govers et al. [37] presented Prompt-GAN, an adversarial method that tunes prompts. Their approach produces both hateful and nonhateful speech texts. Compared to fine-tuning, Prompt-GAN’s architecture reduces the requirements for memory and runtime. Their model improved hate speech categorization F1 scores by up to 10.1%. The Prompt-GAN architecture is composed of the prompt and vocabulary generator, the GPT2/Neo text generation module, and the discriminator network, which serves as a policy engine and feeds the input to the prompt generator.


**Multilingual GAN-PLM:**


Even though GAN-PLM demonstrated remarkable proficiency in generating and learning textual English content, multilingual GAN-PLM expands this ability to encompass many other languages. The integration of multilingualism promotes crosscultural understanding and communication on a worldwide basis, thus gaining benefit from multilingual PLMs (like mBERT) or PLMs that have been pretrained on a specific language (like ChouBERT in French, among others). In 2022, Muttaraju et al. [38] introduced a new approach for binary classification of humorous code-mixed Hindi–English data. Their model outperformed several methods in code-mixed data classification. They investigated the fine-tuned HinglishBERT model into GAN, which gave the best overall results, along with the use of other PLMs such as IndicBERT, MuRIL, and HingBERT within GAN. In 2023, Lora et al. [39] proposed a transformer-based generative adversarial technique for sarcasm detection in Bengali based on Bangla-BERT. They gathered both sarcastic and nonsarcastic comments from newspapers and YouTube and manually annotated them in order to create a dataset. Moreover, Jiang et al. [40] used CamemBERT and ChouBERT in order to construct generative adversarial models. They worked on exploring varied losses over modifying the number of annotated and nonannotated samples in several French datasets to provide a more significant understanding of how to train GAN-BERT models for domain-specific document categorization.

### 2.3. GAN-PLM for Hate Speech Detection

Unlike traditional approaches that depend only on PLMs, Generative Adversarial Networks with Pretrained Language Models (GAN-PLMs) offer a new approach to hate speech detection tasks. GAN-PLMs not only include generative capabilities to produce realistic hateful samples, but they can also identify hate speech patterns in several languages using multilingual PLMs. Through the incorporated use of pretrained language models and generative adversarial networks, GAN-PLMs improve the detection of hate speech while enabling inclusive and cultural sentiment analysis approaches.

In fact, in 2022, Tanvir et al. [26] used a GAN-BERT model based on Bangla-BERT to examine both hate speech and fake news detection in Bengali. They compared the model’s performance to a baseline Bangla-BERT model in order to illustrate the advantages of GAN integration, especially when data samples are scarce. They found that, even with minimally annotated data, their GAN Bangla-BERT model delivered significantly good performance. The experimental results demonstrate how their model outperformed both Bangla-Electra and Bangla-BERT, thereby revealing the importance of incorporating GAN within PLMs. Moreover, using GAN-BERT, Ta et al. [41] developed a method for the Detection of Aggressive and Violent INCIdents in Spanish (DA-VINCIS). As part of a back translation data augmentation technique, they used Helsinki Marian models in order to translate Spanish tweets into English, French, German, and Italian. With each tweet, this technique yields two new texts: the translated text and its back translation. This approach effectively balanced the dataset and reduced the deficiencies in the violent labels when it was specifically applied to the training set across all violent samples. In addition, an ensemble of two semisupervised models was introduced by Santos et al. [42] with the aim to automatically produce a Portuguese hate speech dataset while mitigating bias. The first model combines a GAN-BERT network with Multilingual BERT (mBERT) and BERTimbau, while the second model uses label propagation in order to extend labels from existing annotated datasets to unlabeled ones. With the use of unlabeled data, the GAN-BERT-based approach seeks to modify the label distribution for annotated data. Contrarily, the second approach, based on label propagation, uses dataset samples’ similarities to extend labels to the unlabeled data points.

In 2023, Su et al. [43] introduced SSL-GAN-RoBERTa, a semisupervised model for social media Anti-Asian COVID-19 hate speech detection. Using RoBERTa as the base model, their approach learned from several heterogeneous datasets and enhanced performance accordingly by generating unlabeled data. Their model delivered significant progress in performance over the RoBERTa baseline. Overall, SSL-GAN-RoBERTa learns Anti-Asian speech features from unlabeled samples by employing semisupervised learning-based generative adversarial network technique. Furthermore, the authors managed to show that SSL-GAN-RoBERTa maintains decreased computational costs while outperforming crossdomain transfer learning approaches. Lastly, our previous work, which is a shorter version of the current study [27], presented an innovative approach based on GAN and mBERT to construct a multilingual semisupervised model. With just 20% of the labeled data, we managed to detect hate speech in Indo-European languages. We investigated linguistic feature sharing among these languages and demonstrated its importance for improving sentiment analysis text classification tasks.

Overall, these previous studies have proved remarkable effectiveness, particularly in non-English and many low-resource languages. Researchers have concentrated their efforts on exploring hate speech and offensive language detection in languages like Spanish, Bengali, Portuguese, German, etc., thus constructing customized BERT-based generative adversarial model variations (based on ChouBERT, BanglaBERT, etc.) that are optimized for these linguistic settings. Mostly employed on monolingual techniques, these studies have underlined how adaptable GAN-BERT frameworks are to different linguistic features in the domain. However, the utility of these previous studies is not restricted to monolingual scenarios. In fact, there is a huge trend for utilizing such techniques in multilingual hate speech and offensive language detection, thus emphasizing the pivotal role of generative AI in promoting multilingual and crosslingual analyses. Therefore, the objective of our research paper is to develop an innovative solution in the field, a multilingual and zero-shot crosslingual PLM-based semisupervised generative adversarial model. With the use of both unlabeled and labeled datasets, this approach simultaneously trains a mixture of languages, such as Hindi, English, and German, thus enabling linguistic feature sharing across Indo-European languages. Our paper aims to enhance and effectively contribute to multilingual sentiment analysis tasks. Our main objective is to explain the role and usefulness of GAN-based networks in this NLP field. We aim to investigate the adaptability of one of the generative AI techniques—Generative Adversarial Networks (GANs)—in a variety of linguistic contexts. We seek to go beyond traditional supervised machine learning techniques and study the domain of unlabeled data via a semisupervised approach, which is especially relevant in situations with small or nonexistent annotated data.

## 3. Methodology

### 3.1. Semisupervised Generative Adversarial Network: SS-GAN

The Generative Artificial intelligence (Generative AI) field was ultimately converted by Generative Adversarial Networks (GANs), which brought a novel method for producing synthetic data. GANs were first proposed by Goodfellow et al. in 2014 [24], and they are set on the interchange of two basic parts: the discriminator (D) and the generator (G). These two neural networks are trained against one another in an adversarial context aiming to continually improve the performance in the corresponding task (such as text classification). The generator’s primary role is to generate synthetic data that closely simulates real training data. Yet, the discriminator inspects these produced data samples and distinguishes them from real data. This process goes on iteratively as training runs on until the generator produces more realistic data, and the discriminator gets better at differentiating between real and fake generated samples. The adversarial approach in GANs can be recapitulated by the following equation:$$\min_{G}\max_{D}V\left(D,G\right)=E_{x∼p_{\mathrm{data}}\left(x\right)}\left[\log D\left(x\right)\right]+E_{z∼p_{z}\left(z\right)}\left[\log\left(1-D\left(G\left(z\right)\right)\right)\right]$$
where we define the following:*G*, the generator, minimizes the probability that the discriminator accurately classifies its generated samples as fake.*D*, the discriminator, maximizes its capability to accurately categorize real data as real and generated data as fake.V(D,G) illustrates the value function that both the generator and discriminator aim to optimize through adversarial training.

GANs have been demonstrated to be capable of generating data with complex features and structures, which are similar to real-world datasets. They have shown their adaptability in many fields, from synthesizing realistic images to constructing text and audio. This has extended opportunities for applications in computer vision, NLP, and many other domains [44].

Following the revolutionary work on GANs by Goodfellow et al., there was an interest in investigating various ways to improve and expand upon the original GAN framework. Among these evolutions, semisupervised SS-GANs were introduced by Salimans et al. in 2016 [25], which was a significant turning point in the field. Semisupervised learning in GANs represents a novel case in which the discriminator annotates the data samples in addition to distinguishing between true and fake samples. This helps GANs to be used for semisupervised classification tasks, thus extending their capacities beyond only generating. Compared to separate classifiers and traditional GANs, this hybrid method enables the use of GANs’ adversarial training in both generative and classification tasks simultaneously. Adding to that, SS-GANs effectively employ both labeled and unlabeled data, which is especially valuable in situations where labeled data are not available.

Overall, Table 1 sums up a simple explanation of the roles and related loss functions in mathematical formulas of both SS-GAN’s discriminator D and generator G. First of all, let $p_{real}$ and $p_{g}$ denote the real data and generated data distribution, respectively, let $p(\hat{y}=y|x,y=k+1)$ denote the probability that a sample data *x* is associated with the fake class, and let $p(\hat{y}=y|x,y\in \left(1...k\right))$ denote the probability that *x* is considered real.

### 3.2. SS-GAN-PLM

In our study, we used mBERT and XLM-RoBERTa as PLM models in our generative framework. Starting with a pretrained PLM model, GAN layers were incorporated to execute semisupervised learning. By training on a dataset comprising both labeled and unlabeled samples, the resulting model learns to deliver realistic text representations and yield accurate predictions in text classification tasks. By implementing multilingual pretrained language models like mBERT or XLM-RoBERTa, this integration presents a robust framework for leveraging unlabeled data across multiple languages.

In the task of data classification using Multilingual BERT (mBERT) or XLM-RoBERTa, the model generates a vector representation $\left(h_{CLS},h_{s1},\ldots ,h_{sn},h_{SEP}\right)$, with $h_{CLS}$ serving as the sentence embedding. Enriching this with a Generative Adversarial Network (GAN), we present an adversarial generator *G* and discriminator *D* to improve the classification. In fact, *G* produces synthetic sentence embeddings to imitate real data, while *D* differentiates between real data and those generated embeddings. These synthesized embeddings, alongside PLM embeddings, are later fed into the discriminator for final classification. As illustrated in Figure 1, we combined the GAN architecture on top of mBERT and XLM-RoBERTa by including an adversarial generator G and a discriminator D for final classification.

We utilized a Multilayer Perceptron (MLP) architecture to construct both the generator *G* and the discriminator *D*. Initially, *G* receives a 50-dimensional noise vector and converts it into a synthetic data vector $h_{fake}\in R^{d}$. Afterward, *D* evaluates the realism of $h_{fake}$, along with the representation vectors of real data—labeled and unlabeled for each language—developed by PLMs and denoted as $h_{CLS}$. The final layer of *D* incorporates a softmax activation function, thus giving three vectors of logits corresponding to the three labels in our study: ‘hateful and offensive’, ‘normal’, and ‘is real or fake?’ classes. More specifically, during training, if real data are sampled ($h=h_{CLS}$), D will classify them into the 2 classes of the hateful data (‘hateful and offensive’ or ‘normal’); otherwise, if $h=h_{fake}$, D will classify them into all of the three classes.

**No cost at inference time**: The concept of ‘No cost at inference time’ refers to the efficiency of the model during the inference stage, in which computational resources are optimized. After the GAN model is trained, the generator *G* is no longer employed during this inference phase. Rather, only the PLM model and the discriminator *D* are maintained for our classification task. Therefore, since the generator *G* is no longer engaged, there is no extra computational resource consumption during the inference phase. This procedure assures that the model’s performance during the classification task is kept without any additional resources, which results in more cost-effective and efficient inference [31].

## 4. Experiments and Results

### 4.1. Data: HASOC2019 Indo-European Corpora

In the HASOC (Hate Speech and Offensive Content) track at FIRE 2019, Mandl et al. [45] established comprehensive Indo-European Language corpora for hate speech and offensive content classification, which were extracted from Twitter and Facebook platforms. Their work resulted in the collection of three publicly available datasets (https://hasocfire.github.io/hasoc/2019/ (accessed on 1 September 2023)) in each of the following languages: English, German, and Hindi. These datasets were created in order to contain a various scope of linguistic and cultural contexts, thus enabling robust research in multilingual hate speech and offensive language detection. Particularly, the datasets include 40.82%, 26.63%, and 32.54% of the total training data for English, German, and Hindi, respectively. As for the test set, English contains 34.71%, German 25.59%, and Hindi 39.68% of the total test corpora. For each language, they provide the train and test datasets labeled in three subtasks. In our study, we consider only the first subtask, in which the data is binary labeled into (HOF) Hate and Offensive and (NOT) Non Hate-Offensive. Figure 2 displays the class distribution of each language in the training set.

For our study, we focused on the first subtask of the HASOC2019 track. The training dataset was first split, with 80% (∼11.5 k) going to the Unlabeled dataset (U) and the remaining 20% (∼3 k) going to the Labeled dataset (L). Moreover, we made sure that the actual class distribution was maintained in this division. Our main objective was to demonstrate that using Generative Adversarial Networks (GANs) to train models on small, annotated datasets is effective, thereby reducing the need or the dependency on annotated data. When encountering a deficiency of labeled data, traditional machine learning techniques sometimes fail to achieve a good performance because they might not have sufficient data samples to determine robust features in the data. We reproduced a situation where labeled data were scarce, which is widespread in real-world applications, by splitting the training set into two parts: a smaller amount for annotated data (L) and a larger amount for nonannotated data (U). We also aimed to manage the issue of the imbalanced distribution of data across languages and especially classes. In particular, there was a shortage of data samples for some languages and labels like the class imbalance and small size in German training set in this HASOC2019 corpora, which proposes a severe difficulty for traditional classification models. However, within our SS-GAN-PLM model, we planned to reduce the effect of data imbalance on model performance. Our model can overcome these challenges by employing the capacities of PLMs and Generative Adversarial Networks (GANs) within the semisupervised approach to efficiently learn from both labeled and unlabeled data. In fact, by combining PLMs and GANs into a semisupervised learning framework, our model acquires the ability to effectively learn from both labeled and unlabeled data. More specifically, PLMs serve by providing an understanding of linguistic nuances across the languages we use. On the other side, GANs complement PLMs by enabling data augmentation specifically targeted for the multilingual aspect. GANs help our model to generate synthetic data samples in various languages, thereby extending the diversity and size of the training dataset. This generation process is especially beneficial for addressing data imbalances and enhancing the model’s ability to generalize to unseen languages or linguistic variations (within a zero-shot learning paradigm). Furthermore, the semisupervised learning technique allows our models to leverage the knowledge provided in both labeled and unlabeled data during training. Finally, our model is prepared to effectively address the challenges posed by limited labeled data and data imbalance. This methodology not only improves the model’s robustness but also increases its generalization and relevance to real-world scenarios where labeled data may be scarce or imbalanced.

### 4.2. Experiments and Interpretations

#### 4.2.1. Training Scenarios

We focused on training three models, SS-GAN-mBERT, SS-GAN-XLM (based on XLM-RoBERTa pretrained model), and baseline semisupervised mBERT. After yielding unexpectedly low results from the SS-GAN-XLM model, we considered only the best overall results, thus only displaying its performance on the multilingual training paradigm in our paper. We investigated its function and explained the low results it gave in our analysis. We also considered mBERT as the baseline model because it gave us higher results compared to the XLM-RoBERTa model in our work. The training scenarios were as follows:**Multilingual Training Scenario:** We used all data from the three languages in our dataset, English, German, and Hindi, to train both the SS-GAN-PLM and the baseline semisupervised mBERT models in our multilingual training paradigm. Through the inclusion of crosslinguistic features and patterns, our aim was to utilize the sharing features between languages. We could take advantage of the joint linguistic knowledge that exists inside our multilingual training corpora, which improves our models’ adaptability and generalization among different languages. After training, we utilized HASOC2019 test sets in order to evaluate each model’s performance for each language. Figure 3 offers a more clear vision of this training paradigm.**Zero-Shot Crosslingual Training Scenario:** We employed a crosslingual approach to train our models in the zero-shot scenario. We fine-tuned our models on the English dataset, which is larger than the corpora for the other two languages and has richer linguistic resources. After that, we used a zero-shot learning paradigm to evaluate these models’ performance on the test sets in Hindi and German. Using this technique, we investigated the models’ capacity for crosslingual generalization. Figure 3 presents a more explicit description of the training paradigm.**Monolingual Training Scenario:** For every language in our training data, we fine-tuned our models in the monolingual training paradigm by training and testing the models separately on each language. This method contributes to a richer understanding of model behavior across many linguistic contexts by providing insights into the complexities and difficulties unique to each language.

#### 4.2.2. Models Implementation

Considering the high computational resources employed during the training process, we developed the architecture of GAN to be as simple and accurate as possible. Toward that end, we built the model’s generator as a Multilayer Perceptron (MLP) with one hidden layer. Its role is to generate synthetic data vectors from a given noise vector. In fact, the generator performs by converting noise vectors sampled from a standard normal distribution N(0,1), in which each value is extracted from a distribution with a mean (μ) of 0 and a standard deviation (σ) of 1. This initial conversion transforms the input noise vector, more specifically of size 50 in our structure, into a hidden size vector of 512. Afterward, a 0.2 LeakyReLU activation layer is involved; then, a dropout layer with a rate of 0.1 is included within the generator in order to prevent overfitting and improve the model’s robustness. Overall, this simplified structure promotes efficient consumption of computational resources while enabling the generator to effectively produce synthetic data vectors.

Keeping with the computational resources allocation, the discriminator is alternatively built as another hidden layer Multilayer Perceptron (MLP), thus operating together with the generator. This network has been designed to distinguish between real and fake data samples, as well as to detect hate speech and offensive language for final classification. Equivalent to the generator’s structure, the discriminator begins with a linear layer. Then, a LeakyReLU activation function with a value of 0.2 is incorporated into this layer, alongside a dropout layer with a 0.1 dropout rate. Finally, the output layer of the discriminator consists of class logits that include three outputs: one for each of the two classes “Hate and offensive” and “Normal” class and another output for differentiating between fake and real data samples. Class probabilities are derived by delivering these logits into a softmax activation layer. Overall, our final classification outcome is based on this architectural configuration.

In our process, we leveraged the “BERT-Base Multilingual Cased” model (https://github.com/google-research/bert/blob/master/multilingual.md (accessed on 1 September 2023)). This version of BERT has been trained on 104 languages and has a structure with 12 layers, 12 attention heads, and a hidden size of 768. This version of the model is composed of 110 million parameters, which demonstrates how well it can catch complicated linguistic features. The “Cased” version was chosen because it performs well with languages that use non-Latin alphabets, like Hindi. Adding to that, our selection of the “BERT-Base Multilingual Cased” model was also influenced by the computational resources we had. Compared to bigger, more refined large language models, this model is considered lighter. However, we plan to explore and integrate these large language models into our upcoming work. Moreover, for the multilingual scenario, we also integrated XLM-RoBERTa model, more specifically “xlm-roberta-base” (https://github.com/facebookresearch/fairseq/tree/main/examples/xlmr) (accessed on 1 November 2023), in order to obtain a comparison between the effectiveness of this model along with mBERT model on our SS-GAN framework. This version of XLM-RoBERTa contains 12 layers, 12 attention heads, and a hidden size of 768, which contains 270 M parameters and has been trained on over 100 languages [46]. We intend to study the impact of different pretrained language models on this generative AI method within multilingual hate speech detection tasks.

Moreover, our models have been implemented using Pytorch (https://pytorch.org/) (accessed on 1 September 2023) and trained using a batch size of 32 on Google Colab Pro (https://colab.research.google.com/signup) (accessed on 1 September 2023) (V100 GPU environment with 32 GB of RAM). We set the maximum length variable to 200, and we trained our models on five epochs, with a learning rate of 1 × 10^−5^ and AdamW optimizers for both the discriminator and the generator. We used accuracy, precision, recall, and F1 macro scores as the evaluation metrics to measure our models’ results, which are displayed in Table 2.

#### 4.2.3. Results and Interpretations

Regarding the three training scenarios—monolingual, zero-shot crosslingual, and multilingual—the results in Table 2 demonstrate that SS-GAN-mBERT consistently outperformed the baseline mBERT and SS-GAN-XLM in all the languages. When it comes to enhancing performance in the multilingual training paradigm, SS-GAN-mBERT proved to be a highly efficient solution compared to both monolingual and crosslingual training strategies. More specifically, SS-GAN-mBERT yielded the best results, thereby demonstrating its capability in our semisupervised text classification task. In fact, our investigation shows a 6.5% improvement in accuracy and a 6.4% improvement in the F1 score for hate speech detection tasks in Hindi, over the baseline mBERT model, and a significant rise of about 17% in both the accuracy and F1 macro score compared to SS-GAN-XLM in the same training case. These significant gains highlight the SS-GAN-mBERT’s capacity to develop a deeper understanding of the semantic nuances of languages in hate speech detection task. Even with giving the highest accuracy of about 86% on German data, SS-GAN-XLM output a low performance. This can be explained by various factors. In fact, while XLM-RoBERTa proposes multilingual capabilities, its pretraining might not handle enough the complexities of hate speech detection tasks across the languages used in our experiments. In addition, differences in data quality and linguistic nuances could also affect SS-GAN-XLM’s performance.

Similar improvements were also noticed when zero-shot crosslingual training was employed, which highlights further the effectiveness of SS-GAN-mBERT in various linguistic contexts. This model achieved the highest progress with an approximately 12% increase in the accuracy, precision, recall, and F1 macro scores for the hate detection task in Hindi. This result indicates the model’s strength in transferring knowledge between languages, even in cases when annotated data in the target language is scarce. Nevertheless, it is also important to acknowledge the significant results that both the baseline and SS-GAN-mBERT models within the monolingual scenario achieved, where mBERT indicated an accuracy of approximately 84% for German classification task.

The constant outperformance of the SS-GAN-mBERT model in comparison to the baseline mBERT across all of the three training paradigms highlights the rich influence of adversarial training methods in refining the model’s capacity to distinguish complex and variant linguistic features. More specifically, this outcome became more noticeable within the multilingual training process, thus emphasizing the model’s ability to leverage multilingual corpora effectively. Moreover, regarding the dataset imbalance, we focused on considering F1 macro scores as a robust evaluation metric in our experiments. Thus, comparing the languages output, we can say that our models gained the highest performance in Hindi. This distinction can be related to the larger size of the corresponding dataset. Contrarily, the smaller dataset for German showed lower model performance, as the model may have encountered difficulties in generalizing effectively because of the narrowed exposure to relevant linguistic features and contexts in this language.

To acquire a more detailed interpretation of how our SS-GAN-mBERT model performs better than the baseline mBERT (the second best performing model), we considered analyzing the confusion matrices of the best overall results, which in our case are the multilingual training scenario models tested on Hindi test subset. Figure 4 presents the two confusion matrices of both the baseline mBERT and SS-GAN-mBERT models of this training paradigm.

As we can witness in Figure 4, SS-GAN mBERT achieved higher classification accuracy for both the “Hate and Offensive” (HOF) and “NOT hate and offensive” (NOT) classes compared to the baseline mBERT. Particularly, in the “NOT” class, SS-GAN mBERT reached an approximately 79.03% True Positive Rate (TPR), while baseline mBERT achieved around 66.72%, thus indicating considerable progress in correctly classifying nonoffensive data samples in Hindi. Additionally, SS-GAN mBERT presented a more balanced performance across the two classes, with smaller differences in the TPR between “HOF” and “NOT”, thus presenting improved overall classification accuracy.

## 5. Discussions and Future Directions

### 5.1. Effect of Iterative Labeled Data Increase

Based on the results we obtained, as illustrated in Table 2, we took the best training paradigm, which is multilingual training tested on Hindi, and we reiterated the training of both of the models while progressively increasing the size of the annotated dataset L. We carried a fixed number for the unlabeled dataset U while systematically increasing the number of labeled samples. This technique was essential for evaluating the performance and the scalability of the models under various levels of supervision in our semisupervised approach. Our objective in freezing the number of unlabeled samples was to investigate the influence of the labeled data size on model performance. This enabled us to examine to which extent our models could reach acceptable performance independently of annotated data. We aimed to get closer towards a more unsupervised approach, depending primarily on unlabeled data, thus reducing the need for extensive data annotation. Initiating with a small percentage of labeled data samples and progressively increasing it helped us to observe the learning curve of the models and comprehend their behavior as they were exposed to more labeled data. For more details, we maintained the same size of unlabeled material U, then we started by sampling only 1% of L (which presents very few samples at 29 samples) and then increasing the labeled set size with 5%, 10%, 20%, etc. As we already explained in the previous Section 4.2.3, we considered the F1 macro score metric, along with the accuracy metric values.

Based on Figure 5, we can observe the difference between the baseline and SS-GAN-mBERT models, especially when using the smallest percentage of L data, and even with the use of almost the total amount of labeled data (80–90%), the baseline could not reach the performance of SS-GAN-mBERT. Moreover, even with almost yielding the same accuracy for both models, we can witness the difference in the F1 macro score, where it was evident that SS-GAN-mBERT managed to reach the same performance as the baseline model with a very small amount of labeled data (e.g., we can see the same F1 macro score attained by SS-GAN-mBERT with 1% of L, while the baseline needed more than 6% to reach it). Another aspect to consider is the requirement for labeled data. In fact, in this semisupervised framework (whether within SS-GAN-mBERT or mBERT alone), we can see that with the training unlabeled sets provided U, both of the models did not need a big volume of annotated data. More specifically, as presented in Figure 5, baseline mBERT started giving an F1 macro score and accuracy of more than 0.7 with ∼40% of L, while SS-GAN-mBERT needed only ∼30% to reach this performance; this demonstrates the benefits of implementing semisupervised learning, as it helps to reduce the necessity of data labeling.

Overall, we managed to show through these experiments that the need for annotated instances is reduced when the GAN structure is applied over semispervised mBERT, and it can be reduced more when further improving the structure of GAN, which will be our next step in future work to implement more complex and more advanced GAN structures with more hidden layers in both the generator and the discriminator.

### 5.2. Computational Cost at Inference Time

Considering the cost at inference time, as previously mentioned in Section 3.2, we executed a comprehensive study of the training times of each of the models across the training paradigms (multilingual, crosslingual, and monolingual). Since XLM-RoBERTa has a different number of parameters, it took a very different training time; therefore, we did not consider it in this part of our analysis. Eventually, we found that there was not a considerable difference in training duration between the two models: the baseline mBERT and the SS-GAN-mBERT models. The maximum training time difference marked was about 16 min in one of the training scenarios. This emphasizes the hypothesis that the training time of SS-GAN-mBERT remains essentially similar to that of the baseline model semi-supervised mBERT. This remark indicates that SS-GAN-mBERT offers a viable solution for scenarios where both robustness and training efficiency are critical aspects. More specifically, its efficiency in inference time does not require a large extended training duration. Nevertheless, it is worth noting that this conclusion is related to the simple structure of our GAN’s generator (as an MLP). Therefore, there is a high probability that the time gap could broaden when implementing a more complex generator structure, which can help us to better study the inference time within GANs. Overall, we have a big interest towards this matter because it is crucial to consider the environmental influences of model training, particularly in the context of carbon emissions. Our aim is not restricted to revealing the efficacy of SS-GAN-mBERT but also opening new paths for investigating the environmental aspect, which remains an interesting field for sustainable AI development. While our study did not investigate this aspect in detail, the efficiency of SS-GAN-mBERT could eventually show reduced energy consumption and carbon footprint. Notably, both SS-GAN-mBERT and mBERT demonstrated similar levels of computational resource consumption, thus generally ranging from 4.6 to 5.3 MegaBytes (MBytes) depending on the training scenario and the size of the test set. In the majority of these cases, both models consumed almost equal amounts of resources. This suggests future research for a deeper analysis of resource consumption and measurements, thus taking into consideration the existing tools for CO_2_ energy measurements when training machine learning and large language models [47].

### 5.3. Future Directions

The future direction of this study can be grouped into three domains as follows:

**(1) Generator’s Input:** We have used a constant value of the noise vector of dimensions 50 as the input for our generator in the Generative Adversarial Network (GAN). This option is the optimized value that gave us the best overall results from a comprehensive examination of the initial experimental outcomes associated with concerns of computational efficiency. As a result, we were able to balance between the complexity of the model and the computational resources needed for training. Our goal is to develop procedures that can optimize the generator to select the most appropriate noise vector size for any given dataset. This objective aligns with the idea of improving the adaptability and effectiveness of our GAN framework. An example of our future work for achieving this objective is leveraging Wasserstein GAN, which is a variant famous for its capacity to increase the diversity of generated data samples, thus enabling enhanced stability during training [48]. By incorporating strategies such as Wasserstein GAN into our models, we expect not only to improve the nature of our synthesized data but also to get better generalization capabilities of our model to be able to generate more diverse multilingual data closer to the real ones extracted from social media platforms.

**(2) Data Augmentation:** We aim to decrease the effects of class imbalance, thus leveraging new data augmentation techniques that can be considered as a promising future direction. For instance, we can integrate back translation [41] as one of the solutions, thus taking advantage of its efficacy in various NLP tasks, especially multilingual tasks. In fact, besides our efforts to enhance GAN’s accuracy, we consider improving its data augmentation performance using several techniques, such as Conditional GANs [49]. This strategy has illustrated success in generating high-quality and diverse data samples prepared on specific details to be set as conditions, which could help in further enhancing our hate speech detection tasks.

**(3) Large Language Models (LLMs):** Our future objective opens to accomplishing better generalization abilities by employing advanced multilingual Large Language Models (LLMs) instead of mBERT and XLM-RoBERTa, such as BLOOM, GPT-3, LLaMA2, and Gemma. These LLMs provide richer linguistic features and better contextual understanding, which potentially can enhance the efficacy of our proposed model. Even though the use of such LLMs requires much more computational resources, we intend to mitigate resource limitations gradually. Initially, we plan to start with smaller architectures like GPT-2 and Distil-GPT [50], thus profiting from their language modeling abilities. Moreover, we seek to evaluate the influence of these LLMs within the context of the SS-GAN model. By executing extensive experiments and comparison analyses, we aim to explain and compare the effect of each LLM on the generative capabilities of our model, thereby giving a clear vision for decision making and further advancements.

## 6. Conclusions

In this paper, we have introduced a semisupervised approach, the semisupervised generative adversarial pretrained language models SS-GAN-mBERT and SS-GAN-XLM, which displayed remarkable performance in the field of multilingual and zero-shot crosslingual hate speech and offensive language detection across the English, German, and Hindi languages. Our approach contributes to leveraging semisupervised learning methods to dive into the challenge of data annotation scarcity. The inclusion of Generative AI, which in our case is Generative Adversarial Networks (GANs), managed to improve the efficacy of our approach, thereby demonstrating the benefits of combining semisupervised learning and generative modeling techniques. Our study investigated multilingual textual hate speech detection, which presents important challenges in today’s online communication. By utilizing our SS-GAN-PLM model, we contribute to the proceeding actions in moderating online hate speech content, which is a major sensitive problem widespread in online social media platforms. Previous studies have focused on monolingual hate speech and offensive language detection across languages like Bengali, Portuguese, etc., thus producing specific BERT-based generative adversarial models such as GAN-BanglaBERT for Bengali [26], GAN-bertTimbau for Portuguese [42], SS-GAN-RoBERTa for English [43], etc. However, the relevance of these analyses extends beyond monolingual settings. There is a growing tendency to utilize such techniques for multilingual hate speech detection. Therefore, our paper introduced multilingual and zero-shot crosslingual GAN-PLMs. Our focus was on exploring GANs’ adaptability in various linguistic contexts, thus moving beyond traditional supervised machine learning methods, especially in scenarios with limited annotated data. Exceeding hate speech detection, the importance of our research opens to various generative AI fields, and by constructing upon the foundation of GANs, we propose an adaptable framework that can be further adjusted and extended to address generative tasks across other languages. Overall, our paper also underscores the significance of integrating semisupervised learning and generative modeling techniques along with PLMs in addressing real-world challenges such as hate speech detection.

## Figures and Tables

**Figure 1 entropy-26-00344-f001:**
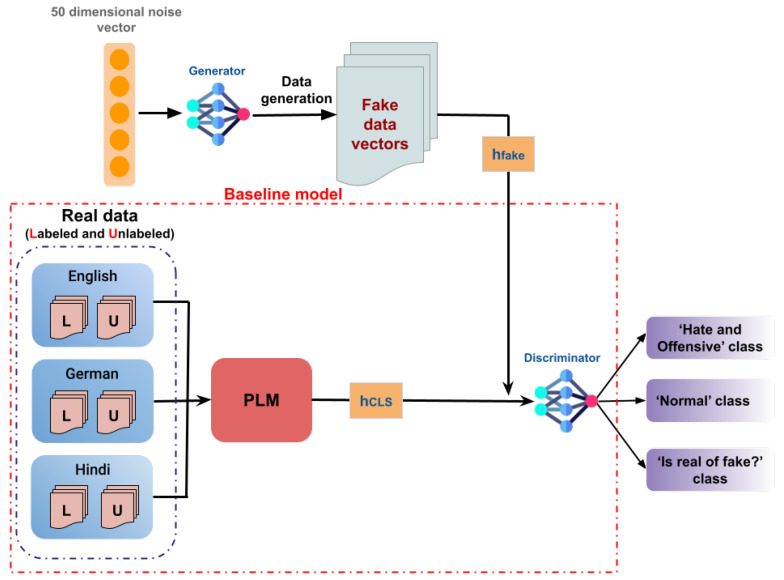
Representation of SS-GAN-PLM architecture for multilingual hate speech detection. [PLM refers to the models we used in our experiments: mBERT and XLM-RoBERTa. “L” denotes the labeled training data, and “U” denotes the unlabeled training data. The process starts with the GAN generator *G* taking a random noise vector as input, which is in our case a 50-dimensional noise vector. *G* then generates synthetic data samples, thus yielding fake vectors $h_{fake}\in R^{d}$. These output samples are fed into the discriminator *D*, alongside the embeddings of both the labeled and unlabeled data processed by the PLM model, which are represented as $h_{CLS}\in R^{d}$ vectors for each language. The discriminator *D* assesses the realism of these inputs, thus distinguishing between real and fake data and simultaneously classifying them into the ‘Hate and Offensive’ and ‘Normal’ classes. This setup enables the training of GAN on both labeled and unlabeled data, thereby leveraging PLM representations to enhance the classification function.

**Figure 2 entropy-26-00344-f002:**
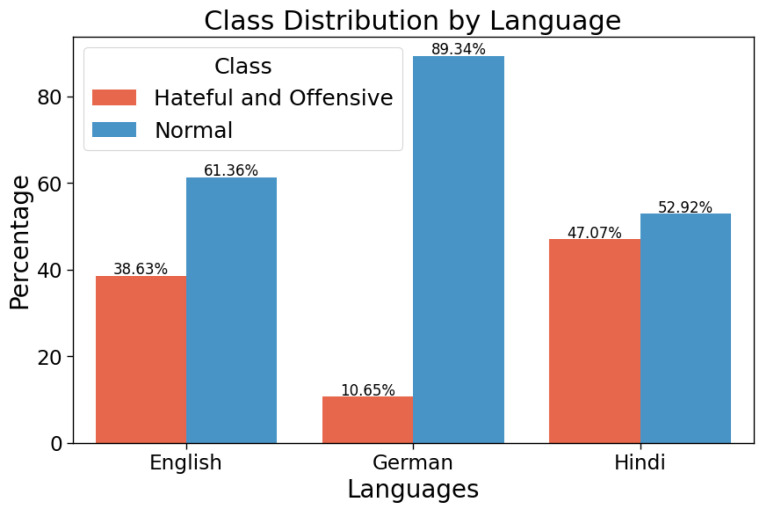
Class distribution variation across languages in the HASOC2019 training dataset. Note: In this corpora, English presents 40.82%, German 26.63%, and Hindi 32.54%.

**Figure 3 entropy-26-00344-f003:**
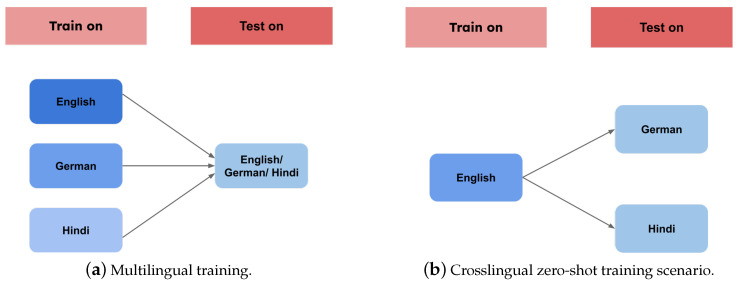
Multilingual and crosslingual training scenarios.

**Figure 4 entropy-26-00344-f004:**
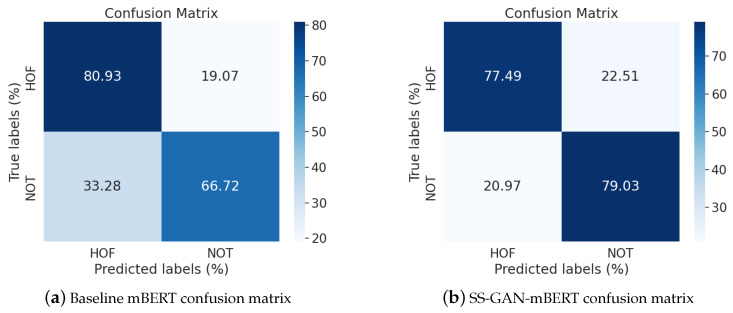
Confusion matrices for mBERT and SS-GAN-mBERT in multilingual training scenario for Hindi.

**Figure 5 entropy-26-00344-f005:**
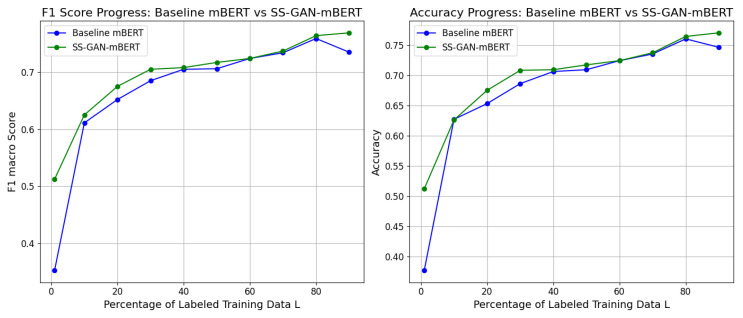
F1 score and accuracy progress on Hindi: baseline mBERT vs SS-GAN-mBERT in multilingual training.

**Table 1 entropy-26-00344-t001:** Roles and loss functions for the discriminator D and generator G in SS-GAN frameworks.

	Discriminator (D)	Generator (G)
**Role **	Training within (k+1) labels, D assigns “real” samples to one of the designated (1,...,k) labels, thus allocating the generated samples to an additional class labeled as k+1.	Generates samples that are similar to the real distribution $p_{real}$ as much as possible.
**Loss Function**	$L=L_{\sup}+L_{\mathrm{unsup}}$ where: $L_{\sup}=-E_{x,y∼p_{\mathrm{real}}}\log\left[p\left(\hat{y}=y|x,y\in \left(1,\ldots ,k\right)\right)\right]$ and $L_{\mathrm{unsup}}=-E_{x,y∼p_{\mathrm{real}}}\log\left[1-p\left(\hat{y}=y|x,y=k+1\right)\right]$ $-E_{x∼G}\log\left[p\left(\hat{y}=y|x,y=k+1\right)\right]$	*L* is the error of correctly identifying fake samples by *D* $L=L_{\mathrm{matching}}+L_{\mathrm{unsup}}$ where: $L_{\mathrm{matching}}={∥E_{x∼p_{\mathrm{real}}}f\left(x\right)-E_{x∼G}f\left(x\right)∥}_{2}^{2}$ and $L_{\mathrm{unsup}}=-E_{x∼G}\log\left[1-p\left(\hat{y}=y|x=k+1\right)\right]$

$L_{\sup}$ is the error in wrongly assigning a label to a real data sample. $L_{\mathrm{unsup}}$ is the error in wrongly assigning a fake label to a real (unlabeled) data sample. f(x) represents the activation or feature representation on an intermediate layer of D. $L_{\mathrm{matching}}$ is the distance between the feature representations of real and generated data.

**Table 2 entropy-26-00344-t002:** Results in monolingual, zero-shot crosslingual, and multilingual training on HASOC2019 dataset.

		English				German				Hindi			
		Acc.	Pr.	Rec.	F1	Acc.	Pr.	Rec.	F1	Acc.	Pr.	Rec.	F1
Monolingual	Baseline mBERT	0.638	0.605	0.629	0.601	0.842	0.489	0.495	0.485	0.696	0.707	0.697	0.693
	SS-GAN-mBERT	0.731	0.668	0.680	0.673	0.811	0.540	0.537	0.538	0.754	0.756	0.755	0.754
Crosslingual	Baseline mBERT	0.638	0.605	0.629	0.601	0.657	0.525	0.551	0.502	0.696	0.707	0.697	0.693
	SS-GAN-mBERT	0.731	0.668	0.680	0.673	0.704	0.568	0.637	0.561	0.754	0.756	0.755	0.754
Multilingual	Baseline mBERT	0.736	0.692	**0.726**	0.699	0.820	0.582	0.585	0.583	0.737	0.743	0.738	0.736
	SS-GAN-mBERT	**0.753**	**0.700**	0.723	**0.708**	0.771	**0.598**	**0.667**	**0.609**	**0.783**	**0.783**	**0.783**	**0.783**
	SS-GAN-XLM	0.686	0.594	0.587	0.590	**0.863**	0.531	0.508	0.495	0.647	0.647	0.647	0.647

In crosslingual training, we used zero-shot learning: training on English and testing on German and Hindi. XLM refers to XLM-RoBERTa model

## Data Availability

Data is contained within the article.

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
