# Peer review of "Multilingual Hate Speech Detection: A Semi-Supervised Generative Adversarial Approach"

_entropy, 2024, doi:10.3390/e26040344_

Round 1
Reviewer 1 Report
Comments and Suggestions for Authors
This study starts with the premise that social media platforms have surpassed cultural and linguistic boundaries, enabling online communication worldwide, intensifying the challenge of online detection of hate speech content.
The authors consider that despite the release of multiple natural language processing (NLP) solutions implementing cutting-edge machine learning techniques, the scarcity of data, especially labeled data, remains a considerable obstacle, which further requires the use of semi-supervised approaches along with Generative Artificial Intelligence (Generative AI) techniques.
Accordingly the authors introduce an innovative approach, a multilingual Semi-Supervised model combining Generative Adversarial Networks (GANs) and Pretrained Language Models (PLMs), more precisely mBERT and XLM-RoBERTa.
The approach proves its effectiveness in the detection of hate speech and offensive language in Indo-European languages (in English, German, and Hindi) employing only 20% annotated data from the HASOC2019 dataset. Presenting significantly high performances in each of multilingual, zero-shot cross-lingual, and monolingual training scenarios. Our study provides a robust mBERT-based semi-supervised GAN model (SS-GAN-mBERT) that outperformed the XLM-RoBERTa-based model (SS-GAN-XLM), and reached an average F1 score boost of 9.23% and an accuracy increase of 5.75% over the baseline semi-supervised mBERT model.
Hate speech detection is a priority for both professionals and researchers: as such any paper advancing in this challenge deserves to be published.
The paper is sound and methodologically correct, however, in its present stage it presents an important flaw that I am sure that the authors will be easily able to fix.
1. RATIONALE
The paper states in the abstract that social media platforms have surpassed cultural and linguistic boundaries, enabling online communication worldwide, intensifying the challenge of online detection of hate speech content.
However, this CRUCIAL topic is not even mentioned in the Introduction that goes straight to technicalities.
So, knowing that this is a technical paper authors should at least introduce the topic to the readership.
In addition, this paper could be interesting for social scientists (which, by the end are the one in charge of analyzing hate speech) so it is of utmost importance to introduce the topic of hate speech on social media. This is key to justify the entire work.
I suggest the authors to start by defining the problem.
There is already a vast literature on this topic.
Likewise there are many examples that have been deeply scrutinized in literature.
Moreover, nowadays social media users implement strategies to avoid censorship.
This paper, for instance, analyzing the Israelo-Palestinian conflict on TikTok, show how activist implement algorithmic resistance, that is to say how they try to avoid censorship manipulating their language.
https://doi.org/10.1080/17513057.2022.2131883
2. DEFINING HATE SPEECH
Even more important: the authors talk about HATE SPEECH without defining it.
This is a serious flaw both theoretically and methodologically because different definititions lead to different legal and technical framework.
This it is seminal to dedicate a section to discuss different definitions of hate speech and clarifying yours.
This paper might help:
https://doi.org/10.1177/2158244020973022
3. CONCLUSIONS
Once the theoretical section has been enriched, in the Conclusions authors should dialogue with existing literature to show how their approach is different.
This will widen the readership and increase the work’s citability.
Good luck!
Comments on the Quality of English Languageno comments
Reviewer 2 Report
Comments and Suggestions for Authors
The paper addressed an important task of HS/OL detection. The authors suggest a new GAN-based approach to cross-lingual and multi-lingual HS detection. The evaluation results demonstrate the validity of this approach.
Detailed comments:
l.241 There are extra dots in this line.
l.356 Because German and Hindi datasets are so imbalanced, would undersampling improve performance or speed up convergence?
l.370-372 Please state exactly how.
l.426 General comment: please distinguish between 'offensive' and 'hate speech' (the latter is stronger in terms of offensive intention). Which one is it? Is this information available? What can you say from manual data examination?
l.554-555 This training time is indeed very plausible.
Round 2
Reviewer 1 Report
Comments and Suggestions for Authors
The paper has improved and is now ready to be published
Comments on the Quality of English Languageno comment